# Decreased Glucocorticoid Receptor Expression and Function in Cord Blood Immune Cells from Preterm Neonates with Morbidity

**DOI:** 10.3390/ijms262110686

**Published:** 2025-11-03

**Authors:** Nana A. O. Anti, Douglas D. Deming, Ciprian P. Gheorghe, Ashra Tugung, Nikia Gray-Hutto, Lubo Zhang, Eugenia Mata-Greenwood

**Affiliations:** 1Lawrence D. Longo MD Center for Perinatal Biology, Department of Basic Sciences, School of Medicine, Loma Linda University, Loma Linda, CA 92354, USA; nanti@students.llu.edu (N.A.O.A.); cgheorghe@llu.edu (C.P.G.); lzhang@llu.edu (L.Z.); 2Department of Pediatrics, Division of Neonatology, Loma Linda University Children’s Hospital, Loma Linda, CA 92354, USA; ddeming@llu.edu; 3Department of Gynecology and Obstetrics, Division of Maternal-Fetal Medicine, Loma Linda University Children’s Hospital, Loma Linda, CA 92354, USA; atugung@llu.edu (A.T.); nhutto@llu.edu (N.G.-H.)

**Keywords:** glucocorticoid sensitivity, neonatal morbidity, cord blood mononuclear cells, white blood cells, lipopolysaccharide, dexamethasone, betamethasone

## Abstract

Glucocorticoids are essential for fetal organ maturation and form the basis of antenatal corticosteroid therapy that has significantly reduced preterm-related morbidity such as respiratory distress syndrome (RDS). However, neonatal morbidity remains a clinical challenge regardless of antenatal corticosteroid therapy. Currently, it is thought that adverse intrauterine environments dysregulate glucocorticoid receptor (GR) homeostasis, yet the biological mechanisms remain poorly understood. Therefore, we aimed to study ex vivo glucocorticoid sensitivity in cord blood immune cells from two independent preterm cohorts to identify associations with neonatal morbidity and uncover potential mechanisms of dysregulated glucocorticoid homeostasis. In the first cohort, thawed cord blood mononuclear cells were exposed to betamethasone in the presence of lipopolysaccharides (LPS) for 4 h. In the second cohort, freshly isolated white blood cells were treated with dexamethasone under unstimulated and LPS-stimulated conditions for 48 h. GR isoform expression and regulation of transactivated and transrepressed genes were assessed via qPCR, immunoblotting, flow cytometry, and ELISA. In both cohorts, reduced GR expression, particularly of the GRα isoform, was observed in neonates with morbidity, but only with culture time and not in freshly isolated cells. Ex vivo impaired glucocorticoid-mediated transrepression of proinflammatory genes *IL6* and *TNF* was also observed in the morbidity groups. In contrast, all samples were comparable in basal immune cell distributions and transactivation of glucocorticoid response element (GRE)-dependent genes *GILZ* and *FKBP5*, irrespective of neonatal morbidity. These findings suggest that neonates that develop morbidities experience an early postnatal GR dysfunction that is potentially programmed in utero. Moreover, under conditions of decreased GR abundance, classical transactivation functions appear to be preserved at the expense of more complex regulatory mechanisms such as transrepression.

## 1. Introduction

Towards late pregnancy, there is a surge in fetal and maternal cortisol levels that play a critical role in fetal development and maturation of a wide range of organs including the lung, brain, heart, kidney, and immune system, which prepare the fetus for extrauterine life [1,2]. In light of this, antenatal corticosteroid therapy (ACS), administered between 28 and 34 weeks of gestation, has become a cornerstone therapy to reduce preterm-related morbidity, such as respiratory distress syndrome (RDS), bronchopulmonary dysplasia (BPD), intraventricular hemorrhage (IVH), and necrotizing enterocolitis (NEC) [3]. However, despite the widespread implementation of ACS, prematurity-related morbidity remains a major clinical concern in the preterm newborn population [4]. Currently, our understanding of the intrauterine environment and genetic factors that regulate endogenous perinatal glucocorticoid sensitivity remain incompletely understood. Preterm neonates are often exposed to a wide array of intrauterine stressors such as hypoxia, infection, growth restriction, and maternal metabolic disease (obesity and diabetes), all of which can impair fetal glucocorticoid signaling and diminish the maturational effects of both endogenous and synthetic glucocorticoids [1], potentially contributing to both increased neonatal complication risk and/or decreased response to perinatal glucocorticoid therapies. Indeed, these same pregnancy complications associated with fetal glucocorticoid dysregulation are also risk factors for neonatal morbidities such as RDS [3,4].

Glucocorticoid effects are primarily mediated by the glucocorticoid receptor (GR) that exerts a wide range of genomic and non-genomic actions [5,6]. GR is structurally diverse, with multiple mRNA splice variants and protein isoforms that modulate tissue-specific glucocorticoid responses [5]. The main active isoform is GRα, which represents 80–90% of all isoforms and is translated into various size proteins due to leaky ribosome shunting [7]. The main active GR protein isoforms are GRα-A, -B, and -C, while GRα-D shows decreased activity and differential gene regulation [5,6]. In addition, GR mRNA isoforms GRβ, GRP, and GRγ are thought to inhibit GRα effects, albeit having specific subcellular activities in a tissue-dependent manner [5,6]. GR genomic actions involve translocation of the liganded receptor into the nucleus, where it regulates gene expression through two main processes: transactivation, which involves GRα homodimerization and binding to DNA consensus glucocorticoid response elements (GREs), and transrepression, which was initially thought to occur primarily through protein–protein interactions with other proinflammatory transcription factors such as NF-κB and AP-1 [5,6]. Recent studies have identified additional GRα monomer-mediated transrepression mechanisms, such as cryptic GRE binding [8,9] and cofactor competition [10], in a gene-specific manner. Collectively, the structural diversity of GR isoforms together with the specific cellular environment underlie the variability in tissue- and context-specific glucocorticoid responses.

Ex vivo peripheral blood mononuclear cells (PBMCs) have been frequently used as an accessible model to study the endogenous glucocorticoid sensitivity in association with disease in a wide range of adult and pediatric diseases [11,12]. In such studies, glucocorticoid sensitivity is typically evaluated by GR expression and/or in vitro functional assays such as gene expression regulation or cellular proliferation. These approaches have shown significant associations between disease and reduced GR expression and/or function [11,12]. Importantly, these studies have also been instrumental in furthering our knowledge on the genetic-, disease-, and physiological-specific determinants of glucocorticoid sensitivity that interact with the progression of the disease as well as with synthetic glucocorticoid response [11,12]. Similarly, GR expression and function have also been investigated in cord blood mononuclear cells (CBMCs) and neonatal PBMCs in association with prematurity and related morbidity. In this field, GR downregulation was found to correlate with RDS severity, with undetectable levels observed in PBMCs, lung, and liver of neonates with fatal disease, and a 57% reduction in those with moderate disease [13,14,15]. Interestingly, reduced GR protein expression was not observed at birth in CBMCs, suggesting that GR downregulation is a consequence rather than a cause of RDS. Studies on CBMCs continue to present equivocal results, with some reporting reduced GRα mRNA levels, while others have found no significant differences [16,17,18]. These studies focused on either GR mRNA or protein levels without assessing GR function, and did not report the severity of RDS [16,17,18]. Furthermore, only Bessler and colleagues have studied ex vivo GR function, and while they demonstrated reduced dexamethasone transrepression of *IL6* and *IL1β* in CBMCs from preterm neonates compared to term neonates [19], the association with neonatal morbidity was not investigated. Overall, further research is needed to fill a critical knowledge gap on the upstream determinants and downstream consequences of perinatal glucocorticoid sensitivity dysfunction in preterm newborns.

In the present study, we examined the ex vivo cord blood immune cell sensitivity to glucocorticoids in association with neonatal morbidity. We hypothesized that prematurity-related morbidity would associate with decreased ex vivo GR expression and function. The rationale that supports this hypothesis is that adverse intrauterine environments, such as placental insufficiency and diabetes, epigenetically dysregulate GR physiology. We employed a comprehensive experimental approach to investigate GR isoform expression at the mRNA and protein levels, as well as GR function, measured as the ability of glucocorticoids to regulate candidate gene transcription and immune cell survival. *FKBP5* and *GILZ* were selected as canonical genes sensitive to glucocorticoid-mediated transactivation, while *IL6*, *TNF*, and *ICAM1* were chosen as representative candidate genes for glucocorticoid-mediated transrepression. In addition, these cytokines have also been implicated in the pathogenesis of neonatal disorders such as RDS and BPD [20,21,22].

## 2. Results

### 2.1. Ex Vivo CBMC Glucocorticoid Sensitivity in Preterm Newborns

We first examined ex vivo glucocorticoid sensitivity using a common approach on thawed mononuclear cells [11,12]. From 32 preterm pregnancies recruited in the original study [23], we analyzed samples from 26 pregnancies, based on availability of sufficient cord blood immune cells for at least two of the following assays: basal protein analysis, basal flow cytometry, and gene expression assays. Maternal and neonatal characteristics are shown in Table 1. Among these neonates, 20 received at least one dose of ACS prior to delivery. Eleven neonates developed one or more preterm-related complications, the majority of which were RDS (9/11 subjects; 3 mild, 3 moderate, and 3 severe). Six of these infants experienced more than one complication. The remaining 15 neonates did not develop any preterm-related morbidity. Maternal and neonatal characteristics were comparable across morbidity groups, including maternal age, maternal BMI, preeclampsia status, mode of delivery, fetal sex, duration between ACS treatment and delivery, and cord blood cortisol levels. However, neonates in the morbidity group had earlier gestational ages and lower birthweights compared to those without complications (Table 1).

#### 2.1.1. Reduced CBMC GR Protein Levels in Preterm Newborns in Association with Morbidity

CBMCs were thawed and rested overnight before analysis of immune cell distribution by flow cytometry. There were no significant differences between the no-morbidity and morbidity groups in immune cell population distribution (Figure 1A). Confirming similar immune cell composition at baseline allowed us to attribute subsequent differences in treatment response to biological effects rather than disparities in starting cell population.

Next, we studied the unstimulated cell expression of GR mRNA isoforms α, γ, P, and β, after 16 h of cell culture to recover from the thawing process (Figure 1B,C). There were comparable GR mRNA isoform levels between morbidity groups, except for GRγ, which was significantly higher by ~94% in the no-morbidity group compared to the morbidity group (Figure 1B). Because of increased GRγ mRNA levels in the no-morbidity group, the abundance of GRα and GRβ were lower than those in the morbidity group (Figure 1C). GRα, as expected, was the most abundant isoform, comprising 82% and 89% of total GR mRNA isoform expression in the no-morbidity and morbidity groups, respectively (*p* < 0.05, Figure 1C). GRγ was the next most abundant isoform (10.2% vs. 5.1% no-morbidity vs. morbidity *p* < 0.05), followed by GRP (7.6 vs. 6.4%, no-morbidity vs. morbidity). GRβ was the least abundant isoform, with only 0.34 and 0.5% in the no-morbidity and morbidity groups, respectively. Furthermore, Western blot analysis of GR protein isoforms revealed five different bands corresponding to GRα-A (~95 kda), GRα-B (~90 kda), GRα-C (~80 kda), GRP (~70 kda), and GRα-D (~50 kda) (Figure 1D). Importantly, CBMCs from preterm neonates with morbidity expressed lower levels of total GR and GRα-A/B protein compared to the no-morbidity group (Figure 1E).

#### 2.1.2. Differential Glucocorticoid and Lipopolysaccharide-Mediated Regulation of CBMC-GR Expression According to Morbidity

Betamethasone dose–response curves were generated in the presence of LPS to study the regulation of GR mRNA isoform expression (Figure 2). Interestingly, LPS significantly upregulated the expression of all GR mRNA isoforms in the no-morbidity group but only that of GRP and GRβ in the morbidity group (Figure 2A–D). Furthermore, we observed a betamethasone dose-dependent downregulation of all isoforms in the no-morbidity group (Figure 2A–D), while there was only significant downregulation of GRP and GRβ in the morbidity group (Figure 2C,D). Differences in GR mRNA expression were more apparent when all LPS-treated samples were pooled (Figure 2E,F). While there were no differences in unstimulated GRα (Figure 1B), LPS induced higher stimulation of GRα in the no-morbidity group, resulting in increased GRα expression compared to the morbidity group (Figure 2E). In contrast, there was a drop in GRα abundance from 89% in unstimulated cells (Figure 1C) to 83.5% in LPS-stimulated cells in the morbidity group (Figure 2F). Furthermore, LPS treatment also increased GRβ expression and abundance in the morbidity group compared to the no-morbidity group (Figure 2E,F). Increased GRγ expression in the no-morbidity group compared to the morbidity group persisted after treatment with LPS and betamethasone (Figure 2E).

#### 2.1.3. Reduced Glucocorticoid-Mediated Transrepression in CBMCs from Neonates with Morbidity

Next, we investigated betamethasone-mediated regulation of both gene transactivation and transrepression. Firstly, we analyzed the glucocorticoid-sensitive genes, *FKBP5* and *GILZ*, which are upregulated via direct GR binding to GRE sites within their promoter and intronic regions. There were no significant differences in the dose-dependent upregulation and EC_50_ values of either *FKBP5* or *GILZ* between morbidity groups (Figure 3A,B). However, betamethasone-mediated *FKBP5* fold activation, but not that of *GILZ*, was significantly higher in the no-morbidity group compared to the morbidity group at both 10 nM and 100 nM doses. Notably, upregulation at the lowest dose of betamethasone was significant only in the no-morbidity group (Figure 3C,D). We then examined betamethasone-mediated transrepression of *IL6*, *TNF*, and *ICAM1*. Significant transrepression of *IL6* and *TNF* was observed in both morbidity groups (Figure 3E,F). However, CBMCs from neonates with morbidity showed significantly lower IC_50_ values than cells derived from the no-morbidity group (Figure 3E,F). Higher IC_50_ values represent decreased glucocorticoid potency. LPS also induced a stronger upregulation of TNF and ICAM1 in the morbidity group, but upregulation was similar for IL6. Surprisingly, betamethasone did not downregulate *ICAM1* expression in any morbidity group (Figure 3G).

### 2.2. Ex Vivo CBWBC Glucocorticoid Sensitivity in Preterm Newborns

We recruited a second cohort to investigate glucocorticoid sensitivity in samples containing neutrophils to fill a critical knowledge gap. A total of 36 women were enrolled, including 8 twin and 1 triplet pregnancies. In all, 46 cord blood samples were expected. However, due to insufficient cord blood sampling and postnatal diagnosis of congenital anomalies, only 40 samples were included in the analysis (Table 2). Unlike the first cohort, which included neonates with a wider gestational age range (25–36 weeks), this cohort was intentionally aimed to recuit older gestational ages (28–34 weeks) to study RDS as a primary outcome. Indeed, of the 40 neonates included, 15 were later diagnosed with RDS, with only 4 of them further developing another neonatal complication (BPD or NEC). RDS severity was mild, and the RDS and no-RDS groups were comparable across several maternal and fetal characteristics including maternal age, pregravid BMI, mode of delivery, ACS treatment-to-delivery interval, fetal sex, birthweight, gestational age, cord blood cortisol levels, and pregnancy complications such as multiple gestation, preeclampsia, maternal diabetes, premature preterm rupture of membranes, and intrauterine growth restriction.

#### 2.2.1. Similar GR Expression in Baseline CBWBCs According to Morbidity

We first examined baseline immune cell frequencies and GR mRNA and protein isoform expression in freshly isolated CBWBCs. Similar to thawed CBMCs, there were no significant differences in main immune cell population frequencies between morbidity groups (Figure 4A). Interestingly, analysis of intracellular GR expression by flow cytometry did not reveal any differences between morbidity groups in total cell levels or according to immune cell subtypes (Figure 4B). Immunoblotting of total basal protein demonstrated four GR protein isoforms, similar to CBMCs: GRα-A/B (~95–90 kda), GRα-C (~80 kda), GRP (~70 kda), and GRα-D (~50 kda) (Figure 4C). Although GRα-C and GRP protein expression tended to be lower in the RDS group, the differences did not reach statistical significance (Figure 4D). Furthermore, there were no significant differences in baseline GR mRNA isoform levels or abundance according to morbidity (Figure 4E,F). Analysis of GR isoform abundance showed once more that GRα was the most predominant splice variant in CBWBCs (86.7% and 83.4% in no-RDS and RDS groups, respectively), followed by GRP (7.7% and 10.3% in no-RDS and RDS groups, respectively), GRγ (4.5% and 5.3% in no-RDS and RDS groups, respectively), and GRβ (1.1 and 0.9% in no-RDS and RDS groups, respectively) (Figure 4F). 

#### 2.2.2. Differential Glucocorticoid-Mediated Regulation of CBWBC-GR Expression According to Morbidity

We then examined the effect of 48h cell culture on GR mRNA isoform expression in comparison with basal (day 0) levels (Figure 5A,B). Overall, cell culture alone stimulated the expression of GRγ and inhibited that of GRβ in both morbidity groups, but only upregulated GRα in the no-RDS group (Figure 5A). This resulted in significantly lower levels of total GR and GRα mRNA in the RDS group compared to the no-RDS group, but no significant differences in expression levels of other isoforms (Figure 5A). The upregulation of GRα expression in the no-RDS group then led to a significant increase in GRα abundance and decreased abundance of GRγ and GRP compared to the RDS group (Figure 5B). Interestingly, LPS did not significantly increase GR mRNA abundance, nor dexamethasone decreased it, with the exception of dexamethasone-mediated decreases in GRα abundance in the presence of LPS in the no-RDS group only (Figure 5C).

A closer inspection across all six treatment groups revealed significantly lower GRα abundance in RDS compared to no-RDS samples, for solvent, LPS, and dexamethasone 0.1 µM-treated samples (Figure 5C). Furthermore, the RDS group had significantly higher abundance of GRP mRNA in all treatments except LPS + dexamethasone at 1 µM (Figure 5C). LPS and dexamethasone treatments had no effect on GR mRNA abundance, except that dexamethasone downregulated GRα in the no-RDS group (Figure 5C, * *p* < 0.05). We further analyzed dexamethasone regulation of intracellular GR protein levels by flow cytometry. While there was a general tendency for dexamethasone to increase GR expression in innate cells (granulocytes and monocytes) from both morbidity groups, significant upregulation was observed only in granulocytes of both morbidity groups and monocytes from the no-RDS group (Figure 5D,E). Interestingly, LPS significantly inhibited the dexamethasone-mediated upregulation of GR expression in granulocytes and monocytes (Figure 5D,E). LPS and dexamethasone did not regulate intracellular GR levels of NK cells and B cells for either group (Figure 5F,I), while LPS induced a significant increase in GR in CD4^+^ T lymphocytes of the RDS group (Figure 5G). Finally, dexamethasone significantly upregulated GR protein levels in CD8^+^ T lymphocytes from the RDS group only, in the absence of LPS. Furthermore, the no-RDS group exhibited significantly lower GR protein levels in CD8^+^ T lymphocytes compared to the RDS group in all treatment conditions except the solvent-treated samples (Figure 5H).

#### 2.2.3. Glucocorticoid Effects in CBWBC Immune Cell Frequencies

Glucocorticoids are well-known to influence the apoptosis and proliferation processes of various immune cell types, both in vitro and in vivo [24]. Therefore, we studied the changes in immune cell frequencies to uncover cell-based responses to glucocorticoid and LPS treatments. Dexamethasone had no effect on the relative frequencies of neutrophils, monocytes, and NK cells for either morbidity group (Figure 6A–C). In contrast, dexamethasone significantly decreased the frequencies of all lymphocytes—CD4^+^ T, CD8^+^ T, and B cells—in both groups, in an LPS-dependent manner (Figure 6D–F). For instance, dexamethasone significantly decreased CD4^+^ T cell frequencies only in the absence of LPS in both morbidity groups (Figure 6D). In contrast, dexamethasone decreased the frequencies of both CD8^+^ and B lymphocytes in the presence of LPS in both groups, and in the absence of LPS only in the no-RDS group (Figure 6E,F). These findings indicate that LPS interacts with dexamethasone in the regulation of immune cell survival and proliferation.

#### 2.2.4. Reduced GR-Mediated Transrepression in CBWBCs from Neonates with RDS

Consistent with the findings in CBMCs (Figure 3), there were no significant differences in dexamethasone-mediated upregulation of *FKBP5* and *GILZ* mRNA in CBWBCs of both morbidity groups, regardless of LPS stimulation (Figure 7A,C). However, there was significantly higher solvent-treated *GILZ* mRNA in the RDS compared to the no-RDS group, while LPS significantly downregulated *GILZ* only in the no-RDS group (Figure 7C). Analysis of dexamethasone-mediated transactivation as percent of control or LPS levels revealed no differences in *GILZ* upregulation between groups (Figure 7D), but showed lower *FKBP5* transactivation in the RDS group compared to the no-RDS group in the presence of LPS (Figure 7B), similar to the findings in CBMCs (Figure 3C).

When assessing dexamethasone-mediated transrepression, we first noted similar LPS-stimulation of *IL6*, *TNF*, and *ICAM1* expression between morbidity groups (Figure 7E–M). Similar to our findings in CBMCs, there was reduced transrepression of *IL6* and *TNFα* mRNA in the RDS group compared to the no-RDS group, but only in the absence of LPS (Figure 7E–J). Interestingly, we found similar transrepression potential among morbidity groups in the presence of LPS (Figure 7E–J). Furthermore, dexamethasone-mediated transrepression of both *IL6* and *TNF* at both the mRNA and protein levels was significantly higher in the presence of LPS (Figure 7F,I), suggesting a potentiating effect of LPS on dexamethasone effects. Analysis of secreted protein revealed no significant differences between morbidity groups for IL6 (Figure 7G). However, in the absence of LPS, the RDS group showed smaller reductions in TNFα protein secretion compared to the no-RDS group, paralleling the differences observed in *TNF* mRNA levels (Figure 7J). Conversely, dexamethasone had no significant effect on either *ICAM1* mRNA expression (Figure 7K,L) or secreted protein (Figure 7M) regardless of LPS stimulation in both the RDS and no-RDS groups. However, RDS samples responded with significantly higher secretion of soluble ICAM1 in comparison with the no-RDS samples when co-treated with LPS and dexamethasone (Figure 7M). These data suggest differential regulation of *ICAM1* compared to *IL6* and *TNF* by both LPS and dexamethasone.

## 3. Discussion

The present study uncovered novel insights into GR physiology in association with neonatal disease. Using a comprehensive experimental design, we examined both GR expression and function for the first time, employing a variety of methods. Consistent findings were observed irrespective of cell culture duration (4 h vs. 48 h) or methodology: neonatal morbidity, in particular mild RDS, associated with reduced ex vivo GRα expression, which in turn correlated with reduced transrepression of *IL6* and *TNF*. Modest reductions in dexamethasone antiproliferative sensitivity were also noted in CD8^+^ T cells and B cells from neonates with mild RDS. In contrast, glucocorticoid sensitivity in terms of transactivation of canonical target genes *FKBP5* and *GILZ* was preserved in all preterm neonates irrespective of morbidity status. Notably, these differences were absent at the time of collection (day 0), but emerged after at least 16 h of cell culture, supporting our rationale that fetal programming of GR—due to adverse intrauterine environments—predisposes the subject for latent reductions in GR expression and function. Subsequent postnatal environmental stressors, combined with prematurity may result in reduced glucocorticoid sensitivity and participate in the development of neonatal morbidities.

GR expression is constitutively maintained, with some tissues, like the lung, immune cells, brain, and liver, showing the highest expression [1]. Therefore, it is quite remarkable that severe RDS is associated with near complete GR downregulation in the lung, as well as the liver and PBMCs [13,14,15]. Although GR deletions can cause RDS, as shown by animal models with targeted GR knockout [25,26,27,28], human studies have determined that GR downregulation does not precede the development of RDS [16,17,18]. Instead, GR downregulation parallels the development of RDS and correlates with its severity [13,14,15]. Consistent with these reports, we did not observe significant differences in GR expression in freshly isolated cells. However, our study is the first to uncover a latent deficiency in GR regulation ex vivo in association with RDS. The reduction in GR expression was modest, which corresponded to a mild RDS phenotype. Importantly, reduced expression was significant for GRα—the principal biologically active and most abundant isoform mediating the majority of glucocorticoid effects. Therefore, we are proposing that adverse intrauterine stressors epigenetically modify the signaling pathways that maintain GR expression, thereby resulting in a blunted response to either LPS stimulation (in thawed CBMCs) or fetal bovine serum exposure (in fresh CBWBCs), resulting in GR decreases. These findings align with the hypothesis of fetal programming of glucocorticoid homeostasis that results in disruption of GR function postnatally, particularly in the presence of what is known as a “second hit”, such as hypoxia or infection. Therefore, these findings highlight the usefulness of ex vivo models to unravel the molecular mechanisms that downregulate GR expression in RDS, the specific intrauterine stressors that precede these events, and the postnatal stressors that induce further GR downregulation.

Although GRα downregulation is the most clinically relevant process, and also the most consistent finding in both cohorts, we also noted novel regulation of other GR mRNA isoforms in association with neonatal morbidity in thawed CBMCs. For instance, the no-morbidity group had significantly higher GRγ but lower GRβ abundance than the morbidity group. In addition, LPS induced a stronger upregulation of all GR mRNA isoforms in the no-morbidity group, while the morbidity group showed significant upregulation of GRP and GRβ only. Furthermore, glucocorticoids downregulated all GR mRNA isoform expressions in the no-morbidity group, but only that of GRP and GRβ in the morbidity group. Therefore, GRα and GRγ regulation by both LPS and betamethasone was blunted in thawed CBMCs from the morbidity group. GRγ is known to regulate mitochondrial homeostasis—a fundamental organelle that is often dysfunctional in neonatal diseases [29,30]. Therefore, reduced immune cell GRγ in neonates can potentially impact inflammatory and oxidative stress-mediated neonatal diseases.

Conversely, and unexpectedly, neither dexamethasone nor LPS regulated GR mRNA isoform expression in cultured CBWBCs. Furthermore, dexamethasone significantly upregulated intracellular GR protein levels in neutrophils and monocytes of the no-RDS group, and that of neutrophils and CD8^+^ T cells in the RDS group, and only in the absence of LPS. Since flow cytometry detects the native form of GR, it is possible that ligand-bound active GR, which dissociates from the chaperone proteins, binds more readily to the antibody than inactive GR, thereby explaining how dexamethasone paradoxically increased GR protein detection by flow cytometry. Of interest, these dataset highlights the interaction of LPS with dexamethasone in regulating GR expression. Moreover, upregulation of neutrophil GR protein levels observed in the no-RDS group could partly mediate the increased transrepression of *IL6* and *TNF* as discussed below. Therefore, our data show for the first time a potential new role of neutrophils in differential glucocorticoid sensitivity in association with neonatal disease.

Perhaps the most novel and remarkable observation from these studies is that glucocorticoid-mediated transactivation of canonical targets such as *FKBP5* and *GILZ* was preserved whereas transrepression of proinflammatory cytokines (*IL6* and *TNF*) was modestly impaired in association with neonatal disease. Although the specific culture conditions in which this impairment is most apparent differ according to cell type (CBMC vs. CBWBC), the consistent reduction in transrepression observed across independent cohorts and experimental approaches underscores the potential contribution of this mechanism to the pathophysiology of RDS.

The translational significance of reduced transrepression in the development of neonatal pulmonary morbidity is potentially dichotomous. Firstly, reduced glucocorticoid transrepression of inflammatory signals and effectors can result in unchecked inflammation and delayed resolution. Indeed, Jonsson et al. observed a two-fold increase in tracheal aspirate IL6 and TNFα levels between day 2 and 5 in uncomplicated RDS [21]. A 30% reduction in glucocorticoid-mediated transrepression, as observed in the current study, could partially explain the observed rises in these pro-inflammatory effectors. Secondly, reduced transrepression can also impair postnatal organ development, particularly in extremely premature infants, although in vivo models are needed to demonstrate the role of dysregulated GR in postnatal maturational events. During fetal development, transrepression—rather than transactivation—has been shown to be a major mechanism responsible for ACS-induced lung development and prevention of neonatal RDS. Although glucocorticoids stimulate surfactant production through GRE-dependent activation of surfactant proteins., in vivo studies demonstrated that glucocorticoids can prevent RDS in a surfactant-independent manner [31,32]. In contrast, GR-mediated antiproliferative and pro-apoptotic effects on mesenchymal cells, which likely rely on transrepression mechanisms, are critical for proper alveolar development and prevention of respiratory failure, and may continue postnatally [33,34]. Therefore, glucocorticoid-mediated transrepression represents an essential mechanism not only for glucocorticoid anti-inflammatory responses but also for the maturational effects. The decreased transrepression observed in association with mild RDS in this study is therefore a novel and clinically relevant finding that further confirms the importance of GR homeostasis in the development of neonatal lung disease.

Another interesting and pivotal finding was the potentiating effect of LPS on dexamethasone-mediated transrepression of *IL6* and *TNF* in all subjects studied. This effect appears to be independent of GR upregulation, as LPS had no effect on GR mRNA expression, and in fact, downregulated intracellular GR levels in neutrophils and monocytes in both morbidity groups. These data are in accordance with studies from Bessler and colleagues that previously reported heightened dexamethasone-mediated transrepression of CBMC secreted IL-1β and IL-6 protein in the presence of LPS compared to basal conditions [19]. They further showed that preterm and term CBMCs were more sensitive to dexamethasone transrepression than adult PBMCs [19]. Prior animal studies have reported synergistic effects of LPS and glucocorticoid treatment on lung development, including activation of cellular death and senescence pathways [35]. In addition, recent transcriptomic analyses in porcine PBMCs demonstrated that many genes are synergistically regulated by combined LPS and glucocorticoid exposure [36]. It has been proposed that this cooperation underlies the reduced incidence of RDS in preterm newborns exposed to chorioamnionitis, although conflicting data exist and the precise mechanisms remain incompletely understood. Therefore, future research is warranted to uncover the mechanisms responsible for the synergism between LPS and glucocorticoids.

Finally, we uncovered novel regulation of ICAM1 in CBMCs and CBWBCs. While LPS significantly upregulated ICAM1 expression in all samples, glucocorticoids did not significantly downregulate its expression. Although multiple mechanisms of GR-mediated transrepression of ICAM1 have been described in vitro [8,37], ICAM-1 has also been reported as a glucocorticoid-resistant gene in different species and cell types [36,38]. For instance, in human airway smooth muscle cells, dexamethasone inhibited TNFα and IL-1β induction of ICAM-1 protein at 4 h but had no effects after 24 h of incubation. Furthermore, some clinical studies have shown no significant downregulation of soluble ICAM-1 in serum from preterm neonates with BPD, following dexamethasone administration [39]. Therefore, discovery of novel ICAM1 inhibitors might lead to novel therapies in the management of BPD.

### Study Strengths and Limitations

This is the first study utilizing a comprehensive molecular ex vivo approach to evaluate both GR expression and function concomitantly in association with neonatal morbidity, particularly RDS. We utilized two different preterm newborn cohorts and assay designs, which strengthens the robustness of our observations. Additionally, our approach included detailed characterization of GR mRNA and protein isoform expression in association with neonatal morbidity. Furthermore, GR expression and function were studied according to immune cell type, including neutrophils, for the first time. Finally, we investigated both classical (transactivation) and non-classical (transrepression) mechanisms of glucocorticoid-mediated gene regulation using both short-term (4 h) and long-term (48 h) exposures, thereby expanding current knowledge on perinatal glucocorticoid sensitivity.

Some limitations of the study should also be noted. Due to limited cord blood sample volume, CBMCs and CBWBCs were studied in different study populations, and thus caution should be taken when interpreting differences between cell types as there could be individual variability contributing to these differences. Furthermore, the first cohort showed significant differences in gestational age, ACS exposure, and birthweight among morbidity groups, potentially confounding the ex vivo glucocorticoid sensitivity of CBMCs, although previous research has shown that CBMC-GR expression is not regulated by these factors [1]. Lastly, while ex vivo assays provide valuable mechanistic insight, they cannot fully recapitulate the complex in vivo environment influencing glucocorticoid action in the lungs of preterm neonates.

## 4. Materials and Methods

### 4.1. Study Design and Human Subjects

Two different preterm cohorts were used to examine ex vivo glucocorticoid sensitivity in association with neonatal morbidities: the first obtained from a previous case–control study on early-onset preeclampsia with matched preterm controls that generated frozen CMBCs (IRB #5130161), and a second one that yielded fresh CBWBCs (IRB #5100242). For both studies, eligible participants were recruited from the Loma Linda University Children’s Hospital Maternity, and informed consent was obtained prior to delivery. Cord blood was collected at birth for basic science studies, and neonatal clinical data were obtained retrospectively from medical records by a senior neonatologist after completion of in vitro glucocorticoid-sensitivity assays. Researchers were blinded to neonatal outcomes during ex vivo glucocorticoid-sensitivity assays.

Prematurity-related neonatal morbidities were assessed by our neonatologist, Dr. Deming, including RDS, BPD, IVH, NEC, and sepsis. RDS and its severity were diagnosed by initial chest radiographs according to diffuse haziness with air bronchograms [40,41]. BPD and its severity were diagnosed by clinical symptomatology and chest radiographs using the NICHD definition [42]. IVH was diagnosed by routine cranial ultrasounds using Papile’s classification [43]. Necrotizing enterocolitis was diagnosed with clinical symptomatology and abdominal radiographs and staged using Bell’s criteria [44].

#### 4.1.1. Preterm Cohort 1: Frozen CBMCs

The first preterm cohort examined ex vivo glucocorticoid sensitivity using a traditional approach of frozen mononuclear cells exposed to 4 doses of glucocorticoids for a short-time period (4 h) and examining gene expression changes in association with neonatal morbidities including RDS, BPD, and NEC. The original study included 66 pregnancies: 32 preterm and 34 term [22]. The preterm cohort consisted of 16 preterm neonates with an early-onset preeclampsia diagnosis and 16 gestational age-matched controls (10 with premature preterm rupture of membranes, and 6 with spontaneous preterm labor). Inclusion criteria included maternal age of 18–40 years with a singleton pregnancy and a gestational age of 25–36 weeks. Exclusion criteria included multiple gestation, fetal malformations, blood transfusions during pregnancy, illegal drug use, and concomitant pregnancy complications such as gestational diabetes. Cord blood was collected from the umbilical vein immediately after delivery into heparinized tubes. Plasma was separated by centrifugation at 2500 rpm and immediately stored at −80 °C for later analysis. Cord blood mononuclear cells were isolated using Ficoll Paque Plus (Cytiva 17144003, Fisher Scientific, PA, USA) according to the manufacturer’s protocol within 1 h of delivery. Mononuclear cells were immediately frozen and stored in liquid nitrogen for future analysis.

#### 4.1.2. Preterm Cohort 2: Fresh CBWBCs

A second preterm cohort was used to study ex vivo glucocorticoid sensitivity in fresh CBWBCs in order to study the role of neutrophils. Furthermore, to control for confounding factors such as gestational age and ACS exposure, this cohort had specific recruitment parameters such as gestational age range of 28–34 weeks and previous exposure to least one dose of betamethasone (ACS) prior to delivery. The gestational window was chosen to capture the period during which ACS is typically administered to induce fetal organ maturation and prevent neonatal morbidity. Extremely premature newborns (<28 weeks gestation) were excluded in this study because of multiple comorbidities and reduced cord blood sampling. ACS exposure was an important inclusion requisite in order to accurately assess individual endogenous differences in ex vivo glucocorticoid sensitivity. Exclusion criteria included maternal steroid use for asthma or autoimmune diseases, illicit drug use, and congenital disorders. In contrast to the preeclampsia case–control study, and to better reflect the clinical complexity associated with preterm birth and RDS, we included multiple gestation and pregnancies complicated with comorbidities. Cord blood was collected from sterile cups into citrate solution or directly drawn from the umbilical cord vein at the base of the placenta using heparinized syringes for immune cell isolation. CBWBCs were isolated using red blood cell (RBC) lysis buffer (155 mM NH_4_Cl, 10 mM KHCO_3_, 0.1 mM EDTA) at a 1:4 ratio, thoroughly washed, and immediately used for glucocorticoid sensitivity assays. An additional cord blood aliquot was collected to obtain serum, which was stored at –80 °C for future analysis. Basal flow cytometry staining, RNA isolation, and protein isolation were performed immediately following isolation of white blood cells (baseline samples, day 0).

### 4.2. Immune Cell Characterization and Intracellular GR Levels by Flow Cytometry

Immune cell subset distribution was determined using multi-color flow cytometry as previously described [45,46] using a MACSQuant Analyzer 10 Flow Cytometer (Miltenyi Biotec, Bergisch Gladbach, Germany). Fluorescently conjugated antibodies used for immunotypying were obtained from Biolegend, San Diego, CA, USA and include anti-CD16 conjugated with Pacific blue (#302032) for granulocytes, anti-CD14 conjugated with PerCP (Biolegend, CA, USA #301848) for monocytes, anti-CD4 conjugated with APC ( #344614) for CD4^+^ T cells, anti-CD8 conjugated with APC/Cy7 (#344714) for CD8^+^ T cells, anti-CD19 conjugated with PE (#392506) for B cells, and anti-CD56 conjugated with PE/Cy7 (#362510) for NK cells. Anti-CD3 (#300420) was used as a pan-lymphocyte marker. To determine cell viability, we used the FVD506 viability dye (Fisher Scientific, Pittsburg, PA, USA, #65086614). Intracellular GR expression was also determined using the anti-GR antibody (Bio-Rad #MCA2469), diluted in 1X eBioscience permeabilization buffer (Fisher Scientific, Pittsburg, PA, USA, #00-8333-56). Flow cytometry data were analyzed using FlowJo, version 10.8 (BD Biosciences, Franklin Lakes, NJ, USA), following a gating specification previously published [45,46].

### 4.3. Cell Culture

CBMCs were thawed using CTL wash (ImmunoSpot, Cleveland, OH, USA, #CTLW-010) according to manufacturer’s instructions and allowed to rest at 37 °C overnight (14–16 h) in immune cell culture media (RPMI 1640 media supplemented with 10% fetal bovine serum and 1% antibiotic-antimycotic solution). A traditional ex vivo glucocorticoid sensitivity assay based on gene expression regulation was used [11,12], using a 4-Log_10_ dilution of betamethasone (Millipore Sigma, Burlington, MA, USA, #B7005)—10^−6^ to 10^−9^ M—in the presence of 10 μg/mL lipopolysaccharide (LPS) from *E. coli* serotype O55:B5 (Millipore Sigma, Burlington, MA, USA, #L6529) for 4 h. LPS is typically used to generate a pro-inflammatory environment that mimics that of disease [47]. CBMCs (3 × 10^6^) were seeded at a density of 1.5 × 10^6^ per mL in 6-well culture treated plates and harvested for RNA isolation to study glucocorticoid-mediated gene regulation. Dose–response curves were generated in www.mycurvefit.com (accessed on 1 November 2024–30 April 2025), and EC_50_ and IC_50_, indicators of glucocorticoid potency, were determined for each sample. In addition, flow cytometry assays, basal RNA, and protein analyses were performed after the overnight rest to obtain baseline data before betamethasone exposure.

Freshly isolated CBWBCs (3–4 × 10^6^) were cultured in immune cell culture media as described for CBMCs at a density of 1.5–2 × 10^6^ cells/mL with a high (1 μM) and intermediate (0.1 μM) dose of dexamethasone (Calbiotech, El Cajon, CA, USA, #265005) with or without LPS (100 ng/mL) from *E. coli* serotype O55:B5 for 48 h. This time point was chosen to expand glucocorticoid sensitivity assays to glucocorticoid-mediated regulation of both mRNA and protein, as well as changes in immune cell type frequencies as a proxy of survival and proliferation [48]. Cell survival and protein regulation require a minimum exposure of 24 h; however, Bessler et al. [19] found that dexamethasone was not able to decrease basal IL6 of IL1β protein secretion with exposures of 24 h, which led us to choose a longer incubation time. Dexamethasone was selected for use in this arm of the study to enable comparison with the existing literature, as it has been thoroughly validated in ex vivo glucocorticoid sensitivity assays [11,12]. Following the 48 h incubation, cells and supernatants were harvested for RNA isolation, flow cytometry, and cytokine analysis as described below.

### 4.4. RNA Extraction and Real-Time PCR

Total RNA was extracted using TRIzol^TM^ (Fisher Scientific, Pittsburg, PA, USA, #15596026) and purified with the Purelink^TM^ RNA Mini Kit (Fisher Scientific, Pittsburg, PA, USA #12183918). Total RNA (~200 ng for WBCs and ~500 ng for CBMCs) was reverse transcribed using the QuantiTect reverse transcription kit (Qiagen, Germantown, MD, USA, #205313). PCR reactions for various glucocorticoid-response genes were performed in duplicate with a QuantiTect SYBR green qPCR kit (Qiagen, Germantown, MD, USA, #204143). Real-time PCR was run with denaturation at 95 °C for 15 s, annealing at 49–61 °C for 20 s, and extension at 72 °C for 10 s/100 bp. GR isoform mRNA was measured using Taqman probe qPCR with a QuantiTect Probe PCR kit (Qiagen, Germantown, MD, USA, #204343) and various PrimeTime double-quenched probes (IDT), containing a 5′FAM fluorphore, internal ZEN quencher, and 3′ Iowa Black FQ quencher for enhanced specificity and reduced background signal. The Bio-Rad iCycler equipped with real-time optical fluorescent detection system was used for SYBR Green and FAM detection. All primers and probes were designed using the NCBI primer design tool and generated an efficiency between 90 and 110%. All primer sequences, together with their accession numbers, are shown in Appendix A. Negative controls (no template) and positive controls (reference sample) were included in each run to normalize Ct values across plates. Relative gene expression for *IL6*, *ICAM1*, *TNF*, *FKBP5*, and *GILZ* was determined using the delta-delta Ct method with *ACTB* as the reference gene. Relative gene expression for GR mRNA isoforms with respect to *ACTB* was determined using standard curves as previously described [49]. Total GR levels were calculated as the sum of all 4 isoforms and the abundance of each isoform was then calculated. Glucocorticoid sensitivity in gene transactivation was estimated as the percent of basal (unstimulated) levels with the formula 100 * Stimulated (dexamethasone treated)/basal (solvent or LPS). Dexamethasone efficacy in gene transrepression was calculated with the formula 100 × (1 − unhibited levels (LPS or solvent)/dexamethasone treated).

### 4.5. Total Protein Isolation, SDS-PAGE, and Immunoblotting

Protein extracts for immune cells (both CBMCs and CBWBCs) were prepared in urea lysis buffer (8 M urea, 25 mM Tris, 150 mM NaCl, 1 mM EDTA, 1% Triton X-100 at pH 7.5) with 2× protease inhibitors (2 mM phenylmethylsulfoxide, 2 mM sodium orthovanadate, and 2× Halt^TM^ protease inhibitor cocktail, Fisher Scientific, Pittsburg, PA, USA, #87786), sonicated for 20 s at low frequency, and centrifuged at 10,000 rpm × 10 min at 4 °C. Protein samples were denatured with Laemmli buffer containing 100 mM dithiothreitol, separated on 10% sodium dodecyl sulfate polyacrylamide gel (SDS-PAGE) and transferred to polyvinylidene fluoride membranes. Membranes were blocked in 5% non-fat dried milk in 0.05% Tris-buffered saline with 0.1% Tween20 (TBST) for 1 h, and then probed in rabbit mononuclear GR antibody (Abcam, Cambridge, MA, USA, #ab183127) diluted in blocking buffer overnight at 4 °C. To determine the relative abundance of proteins, an internal control (from human umbilical vein endothelial cells) was used in every membrane. After three 10 min washes with TBST, the membranes were incubated with corresponding secondary antibodies diluted at 1:2000. Blots were developed using the SuperSignal™ West Femto Maximum Sensitivity Substrate (Thermo Fisher Scientific, Cat# 34095), imaged on the Azure Biosystems c300 digital imager (Azure Biosystems, Dublin, CA, USA) and quantified using the AlphaView Software (Alpha Innotech, San Leandro, CA, USA). Equal loading was determined by β-actin, and relative abundance across multiple membranes was determined with the internal control as previously described [50].

### 4.6. ELISA Experiments

Cortisol levels were measured in heparinized plasma (thawed CBMC cohort) and serum (fresh CBWBC cohort) samples using a commercially available ELISA kit (R&D systems, Minneapolis, MN, USA, Cat# KGE008B) according to the manufacturer’s instructions. This competitive immunoassay quantifies total cortisol (free and protein-bound) with a reported sensitivity of 0.111 ng/mL. Cross-reactivity is <5% with prednisolone (4.4%), Reichstein’s Substance C (3.4%), and progesterone (1.7%), and <1% with other structurally related steroids, including cortisone, corticosterone, deoxycorticosterone, estradiol, prednisone, and 4-androstene-3,17-dione. All samples were diluted at 1:20, a lower dilution factor than recommended by the manufacturer, to ensure detection within the assay’s dynamic range.

Secreted cytokine concentrations in cell culture supernatants were measured in duplicate using commercially available ELISA kits for human IL-6 (R&D systems, Minneapolis, MN, USA, Cat# DY206; assay range: 9.4–600 pg/mL), human ICAM-1 (R&D systems, Minneapolis, MN, USA, Cat# DY720; assay range: 31.2–2000 pg/mL), and human TNF-α (Biolegend, San Diego, CA, USA, Cat# 430204; assay range: 7.8–500 pg/mL) according to the manufacturer’s protocols. For IL-6 assays, supernatants from LPS-stimulated samples were diluted by 100-fold to ensure detection within the dynamic range. For all ELISA assays, absorbance was measured at 450 nm with 570 nm correction using a Tecan Spark Multimode Microplate reader.

### 4.7. Statistical Analysis

Categorical variables are presented as counts and percentages, and quantitative variables as mean ± standard error (SE). Chi-square and Student’s *t*-tests were used to compare categorical and continuous variables, respectively, between the morbidity groups. All figure data are reported as mean ± SE. Shapiro–Wilk and Levene’s tests were used to assess normality and homogeneity of variances. For one-way ANOVA, composite groups were created based on RDS status and treatment conditions to assess differences across these combined factors. Post hoc analysis was performed using Fisher’s Least Significant Difference (LSD) for equal variances and Tamhane’s T2 for unequal variances. For datasets that were non-normally distributed with unequal variances, the non-parametric Kruskal–Wallis test was applied. A *p*-value < 0.05 was considered statistically significant. All statistics were performed in SPSS version 29.

## 5. Conclusions

Antenatal corticosteroid therapy remains the cornerstone intervention for mitigating preterm-related morbidity, yet RDS and other neonatal morbidities remain a significant society burden. The plethora of pregnancy complications associated with preterm birth represents an important cause of adverse intrauterine environments that alter fetal development and increase the risk of postnatal disease. In this study, we observed an association of neonatal morbidity with ex vivo reduced total GR protein abundance and glucocorticoid-mediated transrepression. Importantly, the preserved transactivation of *FKBP5* and *GILZ* suggests that ligand binding and nuclear translocation capacity of GR is intact, consistent with earlier reports on dexamethasone-binding assays—that specifically detect functional, ligand-ready receptors—showing no significant associations with neonatal morbidity. These findings support a model in which diminished total GR limit maximal receptor signaling under disease conditions. In this model, the canonical receptor function of binding to GRE sites is prioritized over more complex mechanisms of actions such as transrepression when GRα expression is reduced. It can also be speculated that intrauterine pro-inflammatory insults interact with GR signaling, further reducing transrepressive activity that might be needed for organ maturation as well as anti-inflammatory activity. Further research is needed to uncover the molecular mechanisms of GR downregulation and gene-specific transrepression. Furthermore, larger cohorts and studies on specific pregnancy complications such as IUGR, pre-gestational diabetes, and chorioamnionitis are warranted to validate these observations and unravel the mechanisms of dysregulated GR homeostasis according to each pregnancy complication. These future research efforts could eventually lead to informed tailored diagnostic and therapeutic approaches to manage prematurity-related morbidity to improve the management of this vulnerable population.

## Figures and Tables

**Figure 1 ijms-26-10686-f001:**
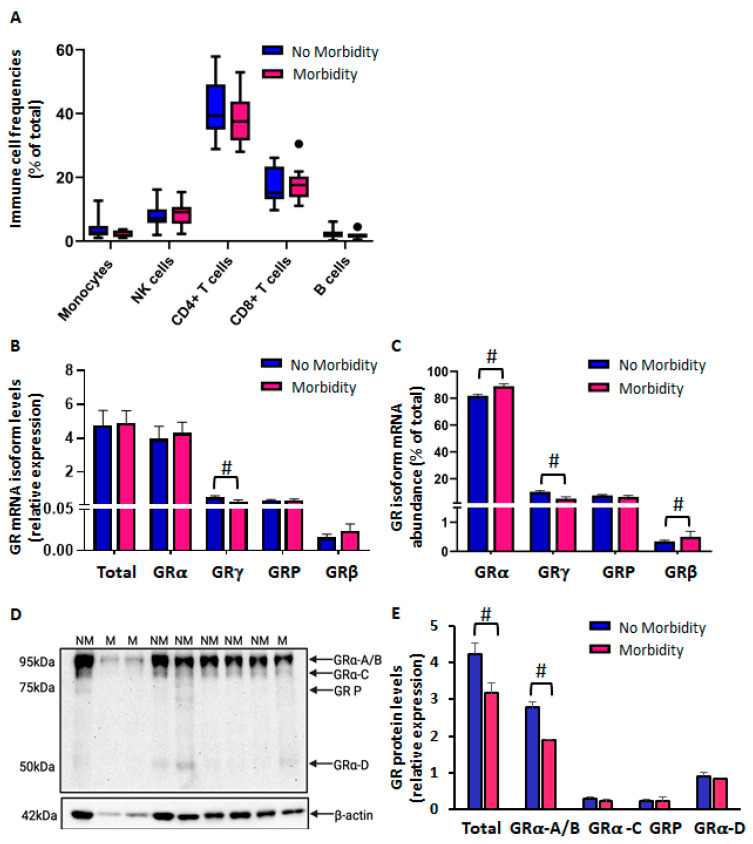
Reduced glucocorticoid receptor expression in thawed CBMCs from neonates with morbidity. Cells were cultured in full media for 16 h before analysis. (**A**) Immune cell subset frequencies measured by flow cytometry (No Morbidity: *n* = 15; Morbidity: *n* = 11). (**B**) Relative expression levels of GR mRNA isoforms (No Morbidity: *n* = 12; Morbidity: *n* = 7). (**C**) GR mRNA isoform abundance shown as % of total GR transcripts (No Morbidity: *n* = 12; Morbidity: *n* = 7). (**D**) Representative immunoblot showing GR isoform protein bands. (**E**) Relative GR protein isoform levels with respect to β-actin expression (No Morbidity, *n* = 14; Morbidity: *n* = 9). Graph bars represent the mean ± SEM. # *p* < 0.05, No Morbidity (NM) vs. morbidity (M).

**Figure 2 ijms-26-10686-f002:**
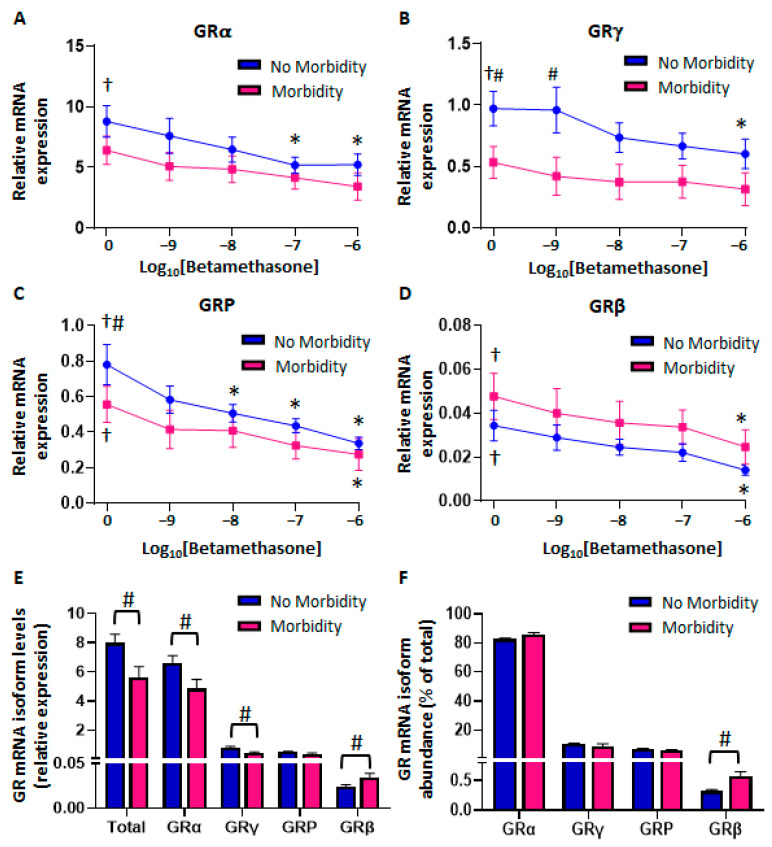
In vitro regulation of GR mRNA expression in CBMCs. Thawed cells were cultured for 16 h to recover from the thawing process, and then treated with LPS (10 µg/mL) and betamethasone (10^−9^–10^−6^ M) for 4 h to generate dose–response curves, shown for (**A**) GRα, (**B**) GRγ, (**C**) GRP, and (**D**) GRβ. Pooled mRNA isoform expression (**E**) and abundance (**F**) across all five treatments (LPS alone and with betamethasone 10^−9^, 10^−8^, 10^−7^, and 10^−6^ M) are shown. Curve lines and graph bars represent the mean ± SEM (No Morbidity (NM): *n* = 12; Morbidity (M): *n* = 7). # *p* < 0.05 NM vs. M, † *p* < 0.05 LPS vs. solvent, * *p* < 0.05 betamethasone vs. LPS only.

**Figure 3 ijms-26-10686-f003:**
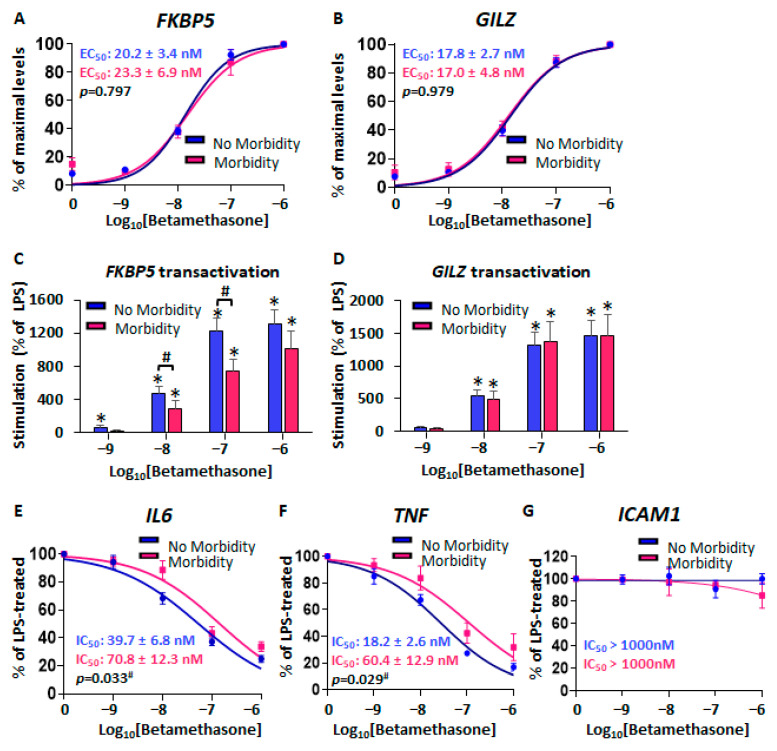
Glucocorticoid regulation of GR target genes in thawed CBMCs. Previously rested cells were cultured in the presence of LPS (10 µg/mL) and betamethasone (10^−9^–10^−6^ M) for 4 h to study GR function. Betamethasone dose-dependent transactivation of *FKBP5* and *GILZ*, shown as percent of maximal levels obtained with highest betamethasone dose (10^−6^ M) (**A**,**B**) and as percent of LPS levels (**C**,**D**). Betamethasone dose-dependent transrepression of LPS-stimulated *IL6*, *TNF*, and *ICAM1* (**E**–**G**). Data represent the mean ± SEM (No Morbidity (NM): *n* = 12; Morbidity (M): *n* = 7). Inserts show the mean ± SEM EC_50_ and IC_50_ for transactivation and transrepression, respectively. # *p* < 0.05 NM vs. M, * *p* < 0.05 betamethasone vs. LPS only.

**Figure 4 ijms-26-10686-f004:**
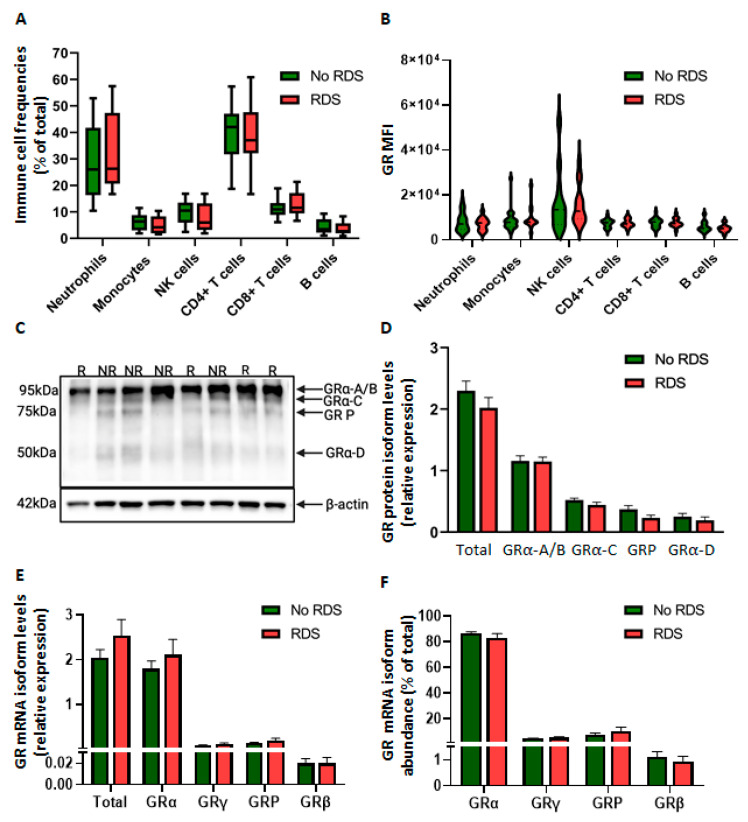
Basal characteristics of fresh CBWBCs. Cells were analyzed immediately after isolation. (**A**) Immune cell subset frequencies as determined by flow cytometry. (**B**) GR protein expression measured by mean fluorescence intensity by immune cell subtype. (**C**) Representative immunoblot and (**D**) relative GR protein isoform levels. (**E**) Relative GR mRNA isoform levels and (**F**) abundance expressed as a percentage of total GR transcripts. Graph bars represent the mean ± SEM (No RDS: *n* = 25; RDS: *n* = 15).

**Figure 5 ijms-26-10686-f005:**
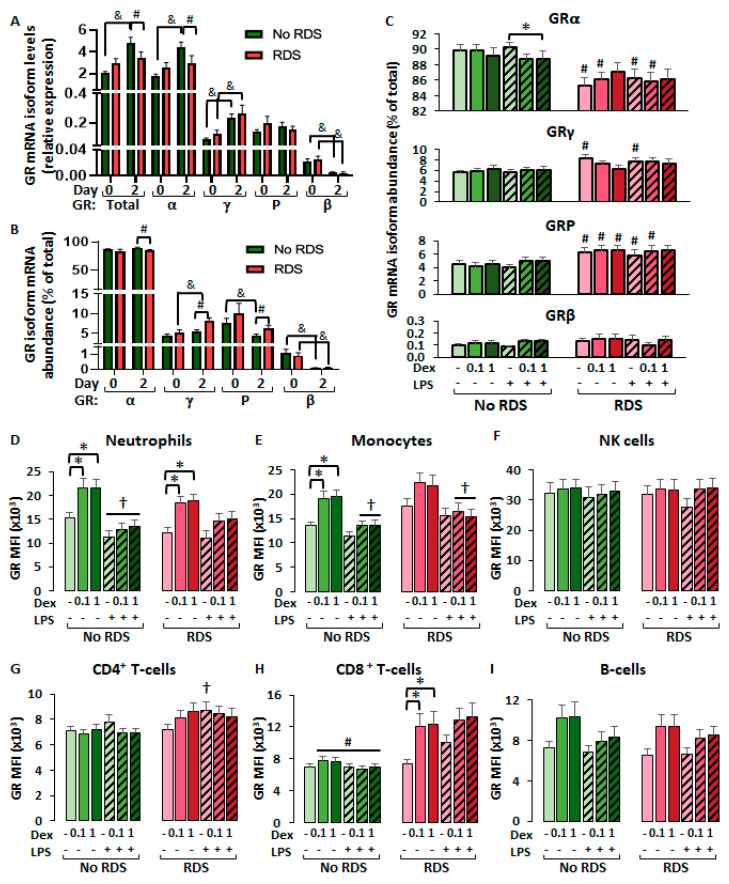
In vitro regulation of GR mRNA expression in CBWBCs. Freshly isolated CBWBCs were treated for 48 h with or without LPS (100 ng/mL), and with or without DEX (0.1 and 1 µM). (**A**,**B**) Baseline (day 0, Figure 4) and solvent-treated (day 2) expression (**A**) and abundance (**B**) of GR mRNA levels. (**C**) GR mRNA abundance across all 6 treatments. (**D**–**I**) Intracellullar GR protein expression across treatment groups and stratified by immune cell subtype: (**D**) CD16^+^ granulocytes, (**E**) CD14^+^ monocytes, (**F**) CD56^+^ NK^+^ cells, (**G**) CD4^+^ T cells, (**H**) CD8^+^ T cells, and (**I**) CD19^+^ B cells. Graph bars represent the mean ± SEM (No RDS: *n* = 25; RDS: *n* = 15). # *p* < 0.05, RDS vs. No-RDS, & *p* < 0.5, day 0 vs. day 2, * *p* < 0.05, DEX vs. no DEX, † *p* < 0.05, LPS vs. no LPS.

**Figure 6 ijms-26-10686-f006:**
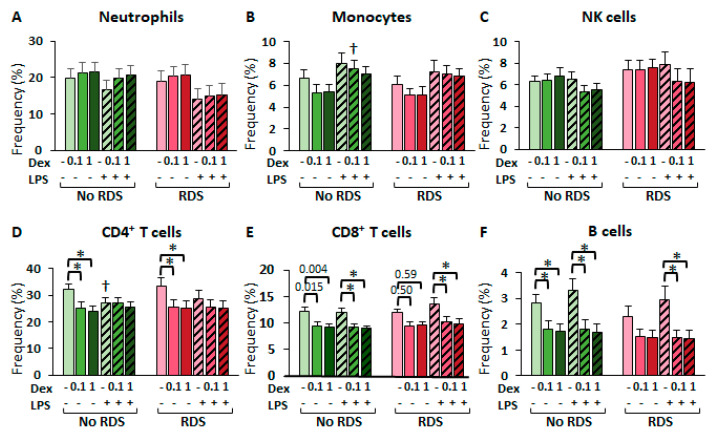
Glucocorticoid regulation of immune cell frequencies in CBWBCs. Freshly isolated CBWBCs were treated for 48 h with or without LPS (100 ng/mL), and with or without DEX (0.1 and 1 µM) and immunophenotyped to indirectly estimate cell survival and proliferation. (**A**) CD16^+^ granulocytes, (**B**) CD14^+^ monocytes, (**C**) CD56^+^ NK cells, (**D**) CD4^+^ T cells, (**E**) CD8^+^ T cells, and (**F**) CD19^+^ B cells. Graph bars represent the mean ± SEM (No RDS: *n* = 25; RDS: *n* = 15). † *p* < 0.05 LPS vs. no LPS, * *p* < 0.05 DEX vs. no DEX.

**Figure 7 ijms-26-10686-f007:**
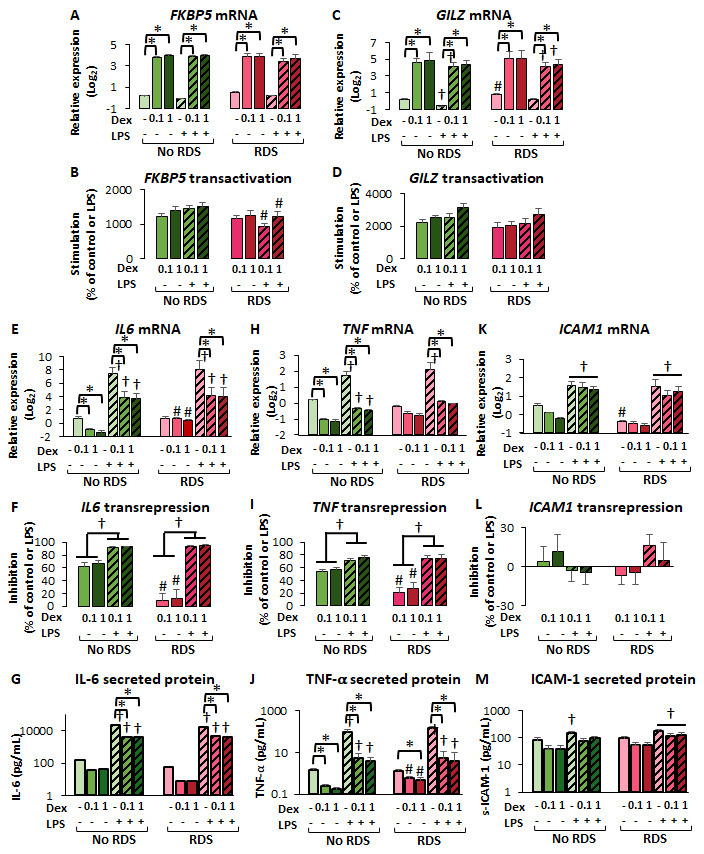
Glucocorticoid regulation of GR target genes in CBWBCs. Freshly isolated CBWBCs were treated for 48 h with or without LPS (100 ng/mL), and with or without DEX (0.1 and 1 µM). DEX-mediated transactivation of GR target genes is shown as relative gene expression and percent transactivation for FKBP5 (**A**,**B**) and GILZ (**C**,**D**). Dexamethasone-mediated transrepression of (**E**) *IL6*, (**H**) *TNF*, and (**K**) *ICAM1* mRNA are shown as log_2_ (relative expression) as well as percent inhibition (**F**–**L**). Secreted protein levels of (**G**) IL-6, (**J**) TNF-α, and (**M**) ICAM-1 were measured by ELISA and are shown as pg/mL. Graph bars represent the mean ± SEM (No RDS: *n* = 25; RDS: *n* = 15). # *p* < 0.05 RDS vs. No-RDS, † *p* < 0.05 LPS vs. no LPS, * *p* < 0.05 DEX vs. no DEX.

**Table 1 ijms-26-10686-t001:** Subject characteristics for thawed CBMC studies.

	Total (*n* = 26)	No Morbidity (*n* = 15)	Morbidity (*n* = 11)	*p*-Value
Maternal characteristics				
Maternal age (years)	28.9 ± 5.7	29.8 ± 1.3	27.8 ± 1.9	0.511
Pre-gravid BMI (kg/m^2^)	26.4 ± 5.2	26.1 ± 1.8	26.9 ± 1.7	0.762
Mode of delivery (C-section, *n*)	20	11	9	0.674
ACS treatment (*n*)	21	10	11	0.046
ACS-to-delivery interval (days)	4.9 ± 1	7.0 ± 3.0	3.0 ± 1.1	0.113
Pregnancy complications				
Preeclampsia (*n*)	12	7	5	1.000
PPROM (*n*)	7	4	3	1.000
Fetal/neonatal characteristics				
Gestational age (weeks)	31.4 ± 6.2	33.3 ± 0.4	28.9 ± 0.9	<0.001
Fetal sex (male, *n*)	16	7	9	0.109
Birthweight (g)	1708 ± 335	2008 ± 115	1297 ± 251	0.022
Birthweight percentile (%)	32.0 ± 6.3	34.9 ± 6.1	28.0 ± 10.2	0.546
Cord blood cortisol levels (ng/mL)	101.2 ± 19.9	97.4 ± 4.8	106.5 ± 5.4	0.223
Neonatal complications (*n*)				
RDS			9	
BPD			4	
NEC			4	
IVH			2	
Neonatal demise			3	

Characteristics are presented as mean ± standard error (SE) for continuous variables and as (*n*) for categorical variables. Statistical significance for continuous variables was determined using the independent Student’s *t*-test, and categorical characteristics were analyzed by the chi-square test.

**Table 2 ijms-26-10686-t002:** Subject characteristics for fresh CBWBC studies.

	Total (*n* = 40)	No RDS (*n* = 25)	RDS (*n* = 15)	*p*
Maternal characteristics				
Maternal age (years)	28.9 ± 0.7	29.2 ± 0.7	28.5 ± 1.3	0.654
Pre-gravid BMI (kg/m^2^)	29.6 ± 1.5	29.2 ± 1.9	30.3 ± 2.2	0.743
Mode of delivery (C-section, *n*)	32	21	11	0.444
ACS exposure (*n*)	40	25	15	1.000
ACS-to-delivery interval (days)	8.1 ± 1.5	8.1 ± 1.9	8.0 ± 2.4	0.964
Pregnancy complications (*n*)				
Maternal diabetes	11	5	6	0.273
PPROM	4	4	0	0.278
Preeclampsia	18	10	8	0.517
IUGR	12	8	4	1.000
Fetal/neonatal characteristics				
Multiple gestation (*n*)	16	10	6	1.000
Gestational age (weeks)	32.5 ± 0.3	32.9 ± 0.3	31.8 ± 0.6	0.118
Fetal sex (male, *n*)	19	10	9	0.328
Birthweight (g)	1819 ± 81	1848 ± 95	1767 ± 153	0.624
Birthweight percentile (%)	27.8 ± 4.9	21.4 ± 23.1	33.7 ± 9.7	0.500
Cord blood cortisol levels (ng/mL)	63.1 ± 4.4	57.4 ± 2.5	72.7 ± 10.8	0.186
Neonatal complications (*n*)				
BPD	1	0	1	0.375
NEC	6	3	3	0.654

Characteristics are presented as mean ± standard error (SE) for continuous variables and as (*n*) for categorical variables. Statistical significance for continuous variables was determined using the independent Student’s *t*-test, and categorical characteristics were analyzed by the chi-square test.

## Data Availability

The original contributions presented in this study are included in the article/Appendix A. Further inquiries can be directed to the corresponding author.

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
