# Peer review of "Decreased Glucocorticoid Receptor Expression and Function in Cord Blood Immune Cells from Preterm Neonates with Morbidity"

_ijms, 2025, doi:10.3390/ijms262110686_

Round 1
Reviewer 1 Report
Comments and Suggestions for Authors
In the manuscript entitled “Decreased Glucocorticoid Receptor Expression and Function in Cord Blood Immune Cells from Preterm Neonates with Morbidity” the authors address the issue of why some preterm babies are unresponsive to glucocorticoids used to treat several preterm issues, including respiratory distress syndrome (RDS). The authors probe the link between expression of the glucocorticoid receptor and its isoforms to responsiveness using cord blood as a proxy for infant GR levels in immune cells. Reduced GR levels in cord blood mononuclear cells (CBMCs) have already been associated with RDS, though the evidence is apparently inconsistent. The authors sought to resolve this controversy using two patient cohorts. In the first, frozen cord blood was thawed and tested for GR levels and glucocorticoid activity (qPCR), and in the second, fresh cord blood was used. The two cohorts were separated into responders (no morbidities) and non-responders (one or more morbidities after glucocorticoid administration), and various comparisons were made.
In general, this is a well-conceived study. The authors state that IL6, TNF, and ICAM1 are “implicated” in that pathogenesis (refs 19-21). It is not clear whether the authors are claiming that a failure of glucocorticoids to repress expression of these cytokines is the cause of this pathogenesis, or more correlative, or a predictor of pathogenesis. As I read it, it was not clear whether the concern was general glucocorticoid insensitivity, which would implicate impaired glucocorticoid-induced development, or specificity insensitivity in blood as a cause of inflammation impairing lung function. It needs to be made clear how this work is a substantial step forward from previous work correlating GR and morbidities.
I have two major comments on the manuscript:
- The correlation between gestational age, weight, and morbidities from the frozen specimens suggests that those are the dominant causes, which clouds any interpretation of GR levels and function from these specimens. The specimens appear to be too different to interpret subtle changes in IL6 and TNF production. Also, the authors state that the dose response of FKBP5 was the same but overall activation was higher (page 6, 178-179) – but don’t show the data for overall activation. The data either needs to be included or the statement removed.
- The correlation between the difference in GR levels and the regulation of downstream genes is not clear. The differences in GR levels, either by qPCR, flow, or blotting, while sometimes significantly different, are not substantially different – a few percent. A change in GR expression of this magnitude is unlikely to change the regulation of genes by much. Also, I don’t understand panels B,D,F, I, and L in Figure 7. For example, the relative expression of FKBP5 in up 3 * log2 with low and high dex compared to control (the first light green bar – is that right? I couldn’t find the doses). Figure 7B states that the stimulation for low and high dex is only 12% compared to control. How can those two things be true? I read the calculation several times, but couldn’t understand it. To my eye, dex gene activation and repression, by qPCR, look to be intact and similar, except IL6 RDS ctl. But the protein secretion, which is more important, looks similar between no RDS and RDS. I can’t see enough of a difference to substantiate the claim that “there was reduced trans repression of IL6 and TNFα in the RDS group compared to the no-RDS group” (line 292, and repeated). The papers cited by the authors claim that there can be 10x as much IL6 in the lungs of preterm babies with morbidities – the subtle differences in the results shown do not approach that. To summarize – the RDS and no RDS for the fresh samples look too similar ot say that there is enough of a difference to cause the morbidity.
Smaller concerns:
- nGREs are a controversial subject. Only the Ortlund group has ever been able to find them, and many have searched their data for them and found no evidence. The mechanism of activation and repression isn’t important to the claims of the paper – I would recommend not wading into that controversy by perpetuating a disputed model.
- Growing cells in culture for 2 days as a model for the early development of cells in babies after birth seems a stretch. What are the culturing conditions – RPMI + 10% FBS (not clear from the methods)? That is not similar to in vivo development. Is there any evidence that cultured cells look like post-birth cells after 2 days?
- The discussion is long. I would recommend sticking to the most solid points. A separate review could be published to encompass the other ideas.
- Tables should have legends that explain the comparisons and statistics, which are not always obvious.
- Figure legends should include concentrations of drugs and a bit more information to help readers follow the information presented.
- Line 225 states, “In the morbidity groups, GRa was significantly more abundant in thawed CBMCs compared to fresh CBWBCs (Figure 1F and 4F).” There is no Figure 1F, and no comparison of relative Gra levels between fresh and frozen.
- The stacked graph is Figure 5C is not easy to read and interpret.
- Line 271 “In contrast, dexamethasone decreased the frequencies of both CD8+ and B lymphocytes in the presence of LPS in both groups, and in the absence of LPS only in the no-RDS group (Figures 6E–F).” The control CD8+ T cell repression by dex looks the same for the RDS and non RDS group. What are the p values for each comparison? Write out the actual p-values to make this claim more substantial – not just a single star and not a star.
Author Response
Reviewer 1:
We thank the reviewer for the thorough assessment of our studies and the key comments raised that helped in improving this manuscript. We have carefully revised the manuscript and corrected it to answer each question. Addition and corrections are shown in red in the revised manuscript and explained below. We hope the revised manuscript will now satisfy the requisitions for acceptance of our work.
1A. In general, this is a well-conceived study. The authors state that IL6, TNF, and ICAM1 are “implicated” in that pathogenesis (refs 19-21). It is not clear whether the authors are claiming that a failure of glucocorticoids to repress expression of these cytokines is the cause of this pathogenesis, or more correlative, or a predictor of pathogenesis.
Answer: The reduction in GR expression and function has important consequences beyond regulation of IL6, and TNF. GR is a master regulator that affects more than 10% of the transcriptome, as well as cytoplasmic , mitochondrial, and membrane signaling pathways of immune cells (references 1, 2, and 36). Therefore, decreased GR expression/function will result in a multitude of dysregulated gene pathways that amplify the biological effects with time. Secondly, and impotantly, RDS is a multifactorial complex disease that involves dysregulation of multiple pathways in addition to that of GR. Lastly, we are not claiming that ‘failure’ of glucocorticoids to repress proinflammatory cytokines is the main cause of RDS or other neonatal morbidities, rather than an important player in the pathophysiology (or development) of the disease. These concepts are better explained in the last 2 paragraphs of the introduction (lines 89-113).
1B. As I read it, it was not clear whether the concern was general glucocorticoid insensitivity, which would implicate impaired glucocorticoid-induced development, or specificity insensitivity in blood as a cause of inflammation impairing lung function.
Answer: Since we studied immune cells, the most direct interpretation of our findings is that there is reduced anti-inflammatory efficacy and potency of glucocorticoids in neonates that develop RDS. Studies on impaired/reduced development (of immune cells o other primary tissues) would require in vitro assays of differentiation, which we did not perform. However, reduced glucocorticoid-mediated differentiation and development due to reduced GR in RDS cannot be discarded in these sick newborns and warrant future studies. This is discussed in lines 388-409. General ‘insensitivity’ described by the reviewer is likely to be observed in fatal severel RDS, since studies (references 13-14) observed undetectable GR levels in lung, liver and fresh PBMCs. In mild RDS cases like the ones in this study, the decreases in GR are modest, as the reviewer stated below, therefore the modest GR dysfunction associates with reduced sensitivity but not general insensitivity.
1C. It needs to be made clear how this work is a substantial step forward from previous work correlating GR and morbidities
Answer: We have modified the introduction (lines 94-98) to explain the novelty and the discussion (lines 317-339) to highlight the significance in terms of moving this field forward as follows:
- This is the first study to use a comprehensive approach to study GR mRNA and protein isoform expression together with GR function/glucocorticoid sensitivity assays. Previous studies focused only on GR protein or GR mRNA, or glucocorticoid transrepression of IL6/IL1/TNF protein levels, and half of the studies did not report the severity of RDS.
- We demonstrate significant downregulation of both GR expression and function in neonates with mild RDS, not only in moderate or severe cases. This finding will open the doors to study the regulation of GR in association with RDS without confounding parameters, such as co-morbidities.
- Decreased GR expression and function is evident in both thawed CBMCs and fresh WBC using different time points, demonstrating reproducibility and providing flexibility for future experimental approaches
- The observed differences occur only after cell culture and are absent at baseline (time 0), supporting the widely-accepted concept of fetal programming of GR dysfunction induced by adverse intrauterine environments.
This last point is particularly valuabel because it supports the use of cord blood cells as an ex vivo model to investigate epigenetic mechanisms in response to specific prenatal stressors. Premature birth is associated with a myriad of pregnancy stressors including placental insufficiency/hypoxia, diabetes/obesity, infection, and prematurity), and currently the role of each stressor, although clinically important, remains incompletely characterized.
2A. The correlation between gestational age, weight, and morbidities from the frozen specimens suggests that those are the dominant causes, which clouds any interpretation of GR levels and function from these specimens.
Answer: We appreciate the reviewer’s comment regarding the potential confounding factors including gestational age and birth weight. We recognize that these parameters often confound studies investigating neonatal pathogenesis. Nevertheless, previous research has not shown a correlation of gestational age, birthweight, and exposure to ACS, with ex vivo PBMC-GR expression (reviewed in reference 1, and now highlighted under limitations in the discussion, line 456). In contrast, RDS is strongly associated with reduced GR expression in multiple tissues, as mentioned above (references 13-17). Lastly, in our own studies, ex vivo GR reductions in expression and function also associated with morbidity in our second separate independent cohort of fresh WBC, where gestational age and birth weight were not significantly different between morbidity groups. Together, these complementary results, aligned with previous reports from other labs, suggest that reduced GR expression and responsiveness are largely driven by other factors independent of gestational age or birthweight, most likely reflecting the effects of exposure to pregnancy stressors (adverse intrauterine environments) and/or RDS pathology.
2B. Also, the authors state that the dose response of FKBP5 was the same but overall activation was higher (page 6, 178-179) – but don’t show the data for overall activation. The data either needs to be included or the statement removed.
Answer: We have included two graphs to show the % transactivation of FKBP5 and GILZ in Figure 3 (panels 3C and 3D). The results are similar to those observed in the second cohort (Figure 7B and 7D): similar GILZ transactivation but lower FKBP5 transactivation by dexamethasone in the presence of LPS in cells from neonates with morbidity.
- The correlation between the difference in GR levels and the regulation of downstream genes is not clear. The differences in GR levels, either by qPCR, flow, or blotting, while sometimes significantly different, are not substantially different – a few percent. A change in GR expression of this magnitude is unlikely to change the regulation of genes by much.
Answer: We appreciate the reviewer’s insightful comment on the translational potential of our findings. Table 1 below gives a quick view of just a few examples of previous studies that demonstrate how a small decrease (or no decrease) in PBMC-GR expression results in significantly decreased ex vivo glucocorticoid sensitivity, in association with clinical resistance to glucocorticoid therapy. In terms of % differences between glucocorticoid sensitive and resistant patients, the degree of change is lowest for ex vivo PBMC GR levels, followed by modest/moderate decreases in ex vivo PBMC glucocorticoid sensitivity and highest decreases in clinical response. The reason that a small reduction in ex vivo GR has a strong impact in the clinic, is because these diseases are complex morbidities with multiple gene pathway dysregulation, and characterized by unchecked inflammation. Therefore, a small decrease in GR has a big impact in a person combating an inflammatory disease. A small difference in GR levels or function ex vivo would then be translated to a difference between no disease and mild disease, or mild disease and severe disease. Therefore, the in vitro results cannot be directly extrapolated, because a given disease cannot be fully reproduced in vitro.
|
Table 1: Decreased ex vivo glucocorticoid sensitivity in the absence of GR expression decreases
|
|||
|
Study subjects |
Δ GR expression ex vivo resistant/sensitive |
In vitro Δ GR function in resistant/sensitive
|
Reference (PMID) |
|
· Adult asthmatics · Glucocorticoid resistant vs sensitive |
· GR α: no difference |
80% lower efficacy in antiproliferative activity |
Goleva et al., 2012 [22236730] |
|
· Asthmatic children · Glucocorticoid resistant (difficult to treat with fluticasone) vs sensitive |
· GR α mRNA 17% reduced |
· 30% lower inhibition of IL8 mRNA (p = 0.02) and TNF mRNA (p = 0.001) |
Goleva et al., 2019 [30059697]
|
|
· COPD patients, healthy smokers, and healthy non-smokers. · COPD = steroid-resistant |
· GR α⁺ cells: no difference · GR α mRNA: no difference |
· 25% reduced dexamethasone-transrepression activity of IFN gamma (p<0.01) |
Kaur et al., 2012 [22417244] |
Compared to these studies, our studies showed significant correlation between GR levels and function in association with RDS. Table 2 summarizes the main results from the second cohort of fresh CBWBCs. The column showing the % deficiency in RDS compared to no-RDS (formula is 1-(RDS/no-RDS)*100) provides the most direct comparison with statistically significant differences highlighted in green. As can be observed, there is correlation between reduced GR expression and reduced GR function in our RDS group. The green highlight means the difference between morbidity groups was significant. The only exception is in IL6 protein secretion where we observe similar results between our RDS and no-RDS groups. This apparent contradictory results is likely due to translation and secretion events that may be GR-independent and thus remain ‘intact’ as the reviewer states in both RDS and no RDS samples.
|
Table 2. Correlation of GR levels with glucocorticoid sensitivity in the absence of LPS
|
|||||
|
Assay |
Method |
No-RDS |
RDS |
% Deficiency in RDS (1-RDS/no-RDS)*100 |
Figure |
|
GR alpha mRNA expression
|
qPCR |
4.56 |
2.99 |
34.4 |
Figure 5A |
|
CD8 cell frequency difference (basal-Dex) |
Flow cytometry |
3 |
2.36 |
21.3 |
Figure 6E |
|
B cell frequency difference (basal-Dex) |
Flow cytometry |
1.1 |
0.78 |
29.1 |
Figure 6F |
|
IL6 mRNA % inhibition (1- Dex/basal *100) |
qPCR |
62.8 |
9.5 |
84.9 |
Figure 7F |
|
IL6 protein secretion % inhibition (1-Dex/Basal *100) |
ELISA |
84 |
74.1 |
11.8 |
Figure 7G
|
|
TNF mRNA % inhibition (1-Dex/Basal *100) |
qPCR |
54 |
20.6 |
61.9 |
Figure 7I
|
|
TNF protein secretion % inhibition (1-Dex/Basal *100) |
ELISA |
83.1 |
55.2 |
33.6 |
Figure 7G |
- Also, I don’t understand panels B,D,F, I, and L in Figure 7. For example, the relative expression of FKBP5 in up 3 * log2 with low and high dex compared to control (the first light green bar – is that right? I couldn’t find the doses). Figure 7B states that the stimulation for low and high dex is only 12% compared to control. How can those two things be true? I read the calculation several times, but couldn’t understand it.
Answer: We thank the reviewer for pointing out the difficulty in understanding these figures. We originally labeled the y axis as: Stimulation (% of ctl or LPS, x 100)). Each number needed to be multiplied by 100. Therefore, the number 12 represents 1200%, not 12%. However, to prevent confusion, we have replotted the panels with the actual percentages on the y-axis. We also included the formulas used to calculate % Inhibition: 100 x [1 – (Glucocorticoid-treated response/maximal stimulated (LPS) or no-inhibition (DMSO)] and % Stimulation: 100 x [Glucocorticoid-treated response/basal or control] in the Methods (lines 583-586).
- To my eye, dex gene activation and repression, by qPCR, look to be intact and similar, except IL6 RDS ctl. But the protein secretion, which is more important, looks similar between no RDS and RDS. I can’t see enough of a difference to substantiate the claim that “there was reduced trans repression of IL6 and TNFα in the RDS group compared to the no-RDS group” (line 292, and repeated). The papers cited by the authors claim that there can be 10x as much IL6 in the lungs of preterm babies with morbidities – the subtle differences in the results shown do not approach that. To summarize – the RDS and no RDS for the fresh samples look too similar ot say that there is enough of a difference to cause the morbidity
Answer: We have partly answered this question above (question 3). The ex vivo differences in GR expression and function are usually magnified in the clinic because of the presence of inflammation and other disease factors. In addition, we are not claiming that GR reductions act alone or that they cause the disease. We are stating that reduced GR expression and function contributes to the development of the disease. And although we studied only 3 proinflammatory genes (IL6, TNF, and ICAM1), GR regulates thousands of genes involved in the inflammatory cascade. Therefore, our results on secreted IL6 protein levels does not minimize or invalidate the significance of GR downregulation. Lastly, the reference that the reviewer alluded to (Jónsson et al, reference 21) is comparing patients with BPD and uncomplicated (mild) RDS, and the results on BPD cannot be extrapolated to our data using mild RDS versus no-RDS (healthy preterm controls). Therefore, the best way to compare our data is with that of the uncomplicated RDS which showed IL6 levels of 250 pg/mL on day 2 that slowly doubled by day 5 and then decreased. These clinical data show that in uncomplicated RDS, IL6 does not rise uncontrollably and is resolved after day 5, which correlates with our data showing similar inhibition of IL6 protein secretion in RDS versus no-RDS. Therefore, we agree with the reviewer that IL6 protein secretion inhibition is similar, but that does not minimize the reduced transrepression observed in IL6 mRNA or that of TNF mRNA and protein. This is now explained in the discussion, lines 388-409.
- nGREs are a controversial subject. Only the Ortlund group has ever been able to find them, and many have searched their data for them and found no evidence. The mechanism of activation and repression isn’t important to the claims of the paper – I would recommend not wading into that controversy by perpetuating a disputed model.
Answer: We agree that the concept of nGREs remains controversial within the glucocorticoid biology field. To avoid controversies, we have deleted the diagrams of figure 3A (canonical transactivation mechanism) as well as 3D (transrepression mechanisms), since we agree with the reviewer that these were not essential for this manuscript.
- Growing cells in culture for 2 days as a model for the early development of cells in babies after birth seems a stretch.
Answer: The experimental approach does not model early development, but studies general GR homeostasis (expression and function). The first cohort used 4h of exposure, which limited our assays to changes in mRNA expression, since glucocorticoid effects in protein and cellular behavior require a minimum of 24h although most researchers use 2-3 days (48-72h). Importantly, Bessler et al (reference 19) studied the effects of dexamethasone on IL6 and TNF alpha protein secretion in the culture media and did not found significant inhibitory effects of dexamethasone on basal levels of IL6 and TNF, suggesting that longer exposures were needed to observe glucocorticoid effects. Therefore, in order to increase the odds of observing glucocorticoid-mediated gene regulation at the protein level and of immune cell survival and proliferation, we selected a 48h exposure. Indeed, we succeeded, as, opposite to Bessler et al, we found significant inhibition of IL6 and TNF in the presence of LPS for all subjects. This rationale is now better explained in the methods, lines 552-557.
What are the culturing conditions – RPMI + 10% FBS (not clear from the methods)? The culturing conditions are described in the Methods (line 537), as RPMI + 10% FBS +1% antibiotic-antimycotic solution.
That is not similar to in vivo development. Is there any evidence that cultured cells look like post-birth cells after 2 days? We would like to emphasize that we did not study in vitro development per se, but ex vivo glucocorticoid sensitivity in terms of gene regulation and cell proliferation/survival. However, primary cells are often used by researchers to study development, maturation and differentiation, and are considered to be a good representation of the individual.
- The discussion is long. I would recommend sticking to the most solid points. A separate review could be published to encompass the other ideas.
Answer: Thank you for this recommendation. We have emphasized the most important concepts, avoided repetition and deleted non-essential components. The discussion is now 15 lines shorter. However, some components are somewhat longer in order to correct and explain some of the other critiques.
- Tables should have legends that explain the comparisons and statistics, which are not always obvious.
Answer: We have revised all table legends to provide clearer descriptions of the comparisons performed and the statistical test used.
- Figure legends should include concentrations of drugs and a bit more information to help readers follow the information presented.
Answer: For Figures 5, 6 and 7 we have changed the x axis to show the actual doses of dexamethasone, to improve the clarity of the graphs and have edited all figure legends to include LPS and dexamethasone concentrations when appropriate.
- Line 225 states, “In the morbidity groups, GRa was significantly more abundant in thawed CBMCs compared to fresh CBWBCs (Figure 1F and 4F).” There is no Figure 1F, and no comparison of relative Gra levels between fresh and frozen.
Answer: thank you for pointing out this mistake. It was Figure 1C and not 1F. However, although we do have stats to back up these results, we decided to delete these sentences because comparisons between thawed MC and fresh WBC on GR isoforms should be done in samples from the same subjects. Therefore, we decided not to compare GR isoforms from the 2 different cohorts, and the 2 sentences were deleted.
- The stacked graph is Figure 5C is not easy to read and interpret.
Answer: To make it clearer, we divided the Fig 5C stacked graph into 4 different narrow graphs to show the abundance of each GR mRNA isoform. In this way, the significant differences between each treatment within morbidity groups is clearer than before.
- Line 271 “In contrast, dexamethasone decreased the frequencies of both CD8+ and B lymphocytes in the presence of LPS in both groups, and in the absence of LPS only in the no-RDS group (Figures 6E–F).” The control CD8+ T cell repression by dex looks the same for the RDS and non RDS group. What are the p values for each comparison? Write out the actual p-values to make this claim more substantial – not just a single star and not a star.
Answer: We completely agree with the reviewer that by the eye, it does not seem like there is any differences between morbidity groups in figure 6E. Indeed, we rechecked the raw data and re-did the SPSS analysis. The p value for the dexamethasone reductions in CD8 T cell population was exactly 0.05 and 0.059 for dexamethasone 0.1 and 1 micromolar doses, respectively in the RDS group. In contrast, the p values for the no-RDS group were 0.015 and 0.004 for dexamethasone 0.1 and 1 micromolar, respectively. This was estimated by 1-way ANOVA as described under methods. Table 2 also shows the differences between morbidity groups, with the RDS goup having a 20% lower reduction in CD8 T cell frequencies compared to the no-RDS. To make this point clearer, we have included the p values just for the effects of dexamethasone on CD8 T lymphocyte frequencies under basal conditions (Figure 6E).

Reviewer 2 Report
Comments and Suggestions for Authors
The manuscript is well-designed, utilizing two cell types and different treatment conditions, with results demonstrating good reproducibility. It systematically reveals the association between complications in preterm infants and decreased GR expression and transcriptional repression function. This suggests that insufficient GRα expression may lead to reduced glucocorticoid anti-inflammatory efficacy in inflammatory conditions, thereby increasing the risk of complications. The study provides a molecular mechanism-based explanation for the differential responses to glucocorticoid therapy. However, after careful review, several issues need to be addressed in the manuscript before it can be accepted.
- The manuscript mentions that there are no intergroup differences in baseline GR expression in fresh CBWBCs, with differences emerging only after culture. It is suggested to discuss whether this "latent GR deficiency" reflects a programmed change that only manifests under inflammatory or stress conditions. This could help explain why some infants develop RDS only after birth.
- LPS combined with dexamethasone treatment enhanced the transcriptional repression effect, but the underlying mechanism remains unclear. It is recommended to discuss potential signaling pathways involved or to identify this as a direction for future research.
- It is recommended to provide rationale in the methods section for selecting the 4-hour and 48-hour time points, as well as the biological significance of these two time points.
- It is recommended to clearly state in the abstract that "impaired transcriptional repression is a novel mechanism for complications in preterm infants" to highlight the significance of the research.
- It is recommended to add the molecular weight of the β-actin protein band below the immunoblot images.
Author Response
Introduction: We thank the reviewer for the careful evaluation of our manuscript and the perceptive suggestions. We have made alterations and additions to clarify the requested concepts and hope these changes will be sufficient to satisfy this reviewer’s concerns.
- The manuscript mentions that there are no intergroup differences in baseline GR expression in fresh CBWBCs, with differences emerging only after culture. It is suggested to discuss whether this "latent GR deficiency" reflects a programmed change that only manifests under inflammatory or stress conditions. This could help explain why some infants develop RDS only after birth.
Answer: Thank you for the insightful suggestion. We agree that the absence of intergroup differences in GR expression at baseline, with reductions observed only after culture, may suggest a latent or stress-inducible deficiency in GR expression. These results highly support the hypothesis that exposure to adverse intrauterine environments can program GR expression and function, leading to what you nominated as ‘latent GR deficiency’. Future research will help us understand the mechanisms of GR programming that will hopefully enable better diagnosis and management of neonatal complications. This concept is included now in the abstract, introduction and more formally in the discussion (lines 340-353).
- LPS combined with dexamethasone treatment enhanced the transcriptional repression effect, but the underlying mechanism remains unclear. It is recommended to discuss potential signaling pathways involved or to identify this as a direction for future research.
Answer: Thank you for this recommendation. We have included a sentence on this topic in lines 426-427.
- It is recommended to provide rationale in the methods section for selecting the 4-hour and 48-hour time points, as well as the biological significance of these two time points.
Answer: We thank the reviewer for this insightful comment. We initially selected the 4-hour treatment for CBMCs based on previous publications on ex vivo immune glucocorticoid sensitivity assays (references 11 and 12 to allow comparison of results to what has been previously published. For the second cohort, we wanted to expand glucocorticoid sensitivity assays beyond transcriptional regulation of gene expression. The glucocorticoid effects in protein and cellular behavior require a minimum of 24h but usually researchers use 2-3 days (48-72h, references 11 and 12). Importantly, Bessler et al (reference 19) studied the effects of dexamethasone on IL6 and TNF alpha protein secretion in vitro with 24h exposure and did not found any effect of dexamethasone on basal levels while inhibiting LPS-levels on at high concentrations of 10 micromolar. Therefore, we selected a 48h exposure to increase the odds of observing glucocorticoid-mediated effects at the protein level and on cell proliferation/survival. Indeed, we succeeded, as, opposite to Bessler et al, we found significant dexamethasone inhibition of IL6 and TNF mRNA and protein secretion in the presence of LPS for all subjects and in the absence of LPS for some subjects (especially without morbidity). This is delineated in lines 552-557.
- It is recommended to clearly state in the abstract that "impaired transcriptional repression is a novel mechanism for complications in preterm infants" to highlight the significance of the research.
Answer: thank you for this recommendation. We have included a paragraph in the discussion that highlights and summarizes the novelty and significance of impaired glucocorticoid-mediated transrepression, specifically in RDS (lines 388-409).
- It is recommended to add the molecular weight of the β-actin protein band below the immunoblot images.
Answer: Thank you for pointing out this omission. The molecular weight of beta-actin has been included in both figures 1D and 4C as 42 kDa.
